# Advancing the study of levels of best practice pre-service teacher education students from Spain: Associations with both positive and negative achievement-related experiences

Antonio Granero-Gallegos[1,2☉]*, Huy P. Phan[3☉], Bing H. Ngu[3☉]

**1** Department of Education, University of Almeria, Almería, Spain, **2** Health Research Centre, University of Almería, Almería, Spain, **3** School of Education, University of New England, Armidale, Australia

☉ These authors contributed equally to this work.
* agranero@ual.es

**Data Availability Statement:** The data come from a research project in which a confidentiality agreement has been signed by the funding

## Abstract

The *study of optimal best practice*, coinciding with a person's 'motivational mindset', is an interesting research inquiry for development. Optimal best practice, in brief, relates to the maximization of a person's state of functioning (e.g., cognitive functioning). Moreover, the nature of optimal best practice is positive and motivational, helping individuals to flourish in different courses of action (e.g., academic performance at school). Several research undertakings, non-experimental in design, have provided clear and consistent evidence to substantiate the existing viewpoints and perspectives of optimal best practice. Our proposed investigation, which involved physical education pre-service teacher students from Spain (*N* = 681), explored one notable focus of inquiry–namely, the formation of optimal best practice and its predictive and explanatory account on future adaptive outcomes. As such, using Likert-scale measures and path analysis techniques, we were able to identify two associative patterns: achievement of optimal best practice is positively accounted for by academic self-concept, optimism, and current best practice and, in contrast, negatively accounted for by pessimism; and that optimal best practice could act as a determinant of academic engagement for effective learning. Such associations are significant, providing relevant information for different teaching and research purposes.

## Introduction

*Optimal best practice* [1–3], or optimal functioning, is an area of research development that attracted scholarly interest, both conceptually [4, 5] and empirically [6, 7]. Julian Fraillon [1] briefly introduced the term 'notional best functioning' in the mid-2000 and subsequently, Phan and his colleagues expanded to include a detailed explanation and theoretical account of what this concept actually entails. Notional best functioning, or optimal best practice, according to Phan and his colleagues [2], refers to the "maximization, or fullest capability, of a person's internal state of functioning" (e.g., cognitive functioning) in a specific subject matter. A

institution. The data underlying the results presented in the study are available from Health Research Centre, University of Almeria (contact: ceinsa@ual.es).

**Funding:** "This research was funded by the ANDALUSIAN PLAN FOR RESEARCH, DEVELOPMENT, AND INNOVATION (PAIDI 2020) OF THE JUNTA DE ANDALUCÍA, grant number P20_00148 (I+D+i research project)".

**Competing interests:** The authors have declared that no competing interests exist.

typical example of optimal physical functioning, say, is a professional football player's indication of his capability to score 25 goals this season. Academically, likewise, a Year 12 student's optimal best in mathematics may entail her success of achieving an A grade at the end of the school term. An interesting question that we may ask, which also forms the basis of our current study, is the following: what causes a person to achieve optimal best in a specific subject matter? This reflective question, importantly, alludes readers to an important theoretical premise, namely, the notation of '$\Delta_{(L_1-L_2)}$', which depicts the relationship between 'actual best', denoted as $L_1$, and 'optimal best', denoted as $L_2$.

Interestingly, in a recent *conceptual analysis* article, Phan and his colleagues [8] proposed a theoretical account, which the authors termed as 'goals of best practice' (i.e., denoted as 'GsBP'). According to the authors, goals of best practice are personal best goals, which may assist to facilitate the successful achievement of optimal best. This conceptualization of GsBP, in tandem with our previous reflective question (i.e., what causes a person to achieve optimal best?), is innovative and provides grounding for us to conceptualize a theoretical model for examination. Our focus of inquiry, involving university students ($N = 681$) in Spain, addresses one important topical theme–namely, an exploration of *different types of antecedents that could explain the cause of optimal best in academic learning*. From existing literatures, we consider three contrasting variables, which may function as antecedents of best practice: academic self-concept, optimism, and pessimism (e.g., academic self-concept → optimal best). Moreover, consistent with previous research development, we also consider the explanatory and predictive power of levels of best practice for examination. For example, does a student's successful achievement of optimal best practice in a specific subject matter assist to improve their academic engagement of the same subject?

One theoretical element that is noteworthy for consideration is the 'impact' of optimal best practice on a person's development. 'Impact of optimal best', for us, is more than just a predictive effect of optimal best practice (e.g., the positive effect of optimal best practice on academic performance in a subject matter). Our rationalization, in this sense, contends that achieving optimal best practice would require some form of a 'motivational mindset'. A motivational mindset is optimistic, serving to inspire, compel, and/or motivate a person to strive for optimal best, where possible. We reason that *consideration* towards achieving optimal best in a context matter would, likewise, help initiate and/or instil a positive and motivational mindset for deliberation. This postulation also places emphasis on the alternative–that a 'demotivational mindset' is pessimistic, serving to negate and/or to weaken one's resolve and self-determination to strive to achieve optimal best. Overall, then, we conceptualize a theoretical model for examination, which seeks to explore and identify different types of antecedents that could impart contrasting effects on the two levels of best practice. Furthermore, as described earlier, a related issue for examination is the explanatory and predictive role of the two levels of best practice.

## Optimal best practice: In brief

The *study of optimal best practice* [1–3], which Fraillon [1] introduces in the mid-2000s, has important educational and non-educational implications for consideration. Optimal best practice largely reflects the characteristics of the *paradigm of positive psychology* [9, 10], such as a person's state of motivation, aspiration, and optimism to succeed in life. One aspect of positive psychology, in brief, relates to the encouragement, promotion, and fostering of proactivity of human agency–for example, a student's proactivity of human agency may entail their experience of flow [11–13] and flourishing [14–16] in a specific subject matter. Optimal best practice, likewise, refers to the perceived *maximization of a person's internal state of functioning* [1, 2]–for example, in terms of academic learning, a secondary school student's optimal cognitive

functioning may entail her successful achievement of obtaining an A grade for Calculus at the end of the school term. In this sense, conceptually and philosophically, one could argue that successful optimal best practice, or optimal functioning, is analogous to and/or is equivalent to personal experience of flourishing [17].

'Sub-optimal' best practice in a subject matter, in contrast to that of optimal best practice, may refer to a person's sub-par experience and/or unsuccessful performance, reflecting their state of pessimism, demotivation, helplessness, negativity, etc. The study of positive psychology [9, 10], likewise, considers a central focus, which explores preventive measures that could negate and/or weaken sub-optimal best practice. For example, in terms of academic learning, an educator may use encouraging feedback to instil resilience, personal resolve, and self-confidence, motivating a person work hard to overcome perceived hardships, difficulties, etc. In terms of our interest and focus of inquiry, we wish to consider the nature of optimal best and how this positive concept would associate with other adaptive and/or psychological outcomes.

**Underlying nature of actual best and optimal best.** In his seminal publication, titled '*Measuring student well-being in the context of Australian schooling: Discussion Paper*', Fraillon [1] theorized that optimal best practice, also known as 'notional best functioning', would intimately associate with another level of best practice, known as 'actual best functioning'. Actual best functioning, in brief, is defined as a person's current and realistic level of best practice in terms of capability, knowledge, skills, and/or performance (e.g., a student's indication of his current ability to solve equations with one unknown, $x$: $x + 10 = -9$). That a person's sense of 'optimization', according to Fraillon's [1] conceptualization, is the result of their progress from actual best functioning to notional best functioning (e.g., the student's indication of his optimal capability to solve equations with two unknowns, $x$ and $y$: $x + y = -9$ and $2x - y = 10$). Phan and his colleagues [e.g., 4, 5, 17] subsequently used Fraillon's [1] conceptualization of the relationship between actual best functioning and notional best functioning to provide a more comprehensive overview, which we review in this section of the article.

In the most recent updated overview of personal best practice, Phan and Ngu [17] contend that the two levels of best practice, known as 'actual best practice' (denoted as '$L_1$') and 'optimal best practice' (denoted as '$L_2$'), are *contextualized* within a particular timepoint, setting, and subject area. Moreover, according to the authors, progressing from $L_1$ to $L_2$, as shown in Fig 1, requires time, effort, and enactment of the *process of human optimization* [1, 5, 18]. In other words, the relationship between $L_1$ and $L_2$, denoted as '$\Delta_{(L1-L2)}$', does not occur spontaneously and/or instantaneously. Rather, different types of psychological processes, psychosocial factors, and educational practices would act to 'optimize' a person's internal state of functioning from Time 1 to Time 2, etc. For example, in a school setting, the implementation of an appropriate instructional design (i.e., an educational practice) [19, 20] could assist to improve a child's learning experience in a subject matter from Time 1 (e.g., beginning of the school term) to Time 2 (e.g., the end of the school term). In a similar vein, non-academically, optimal health functioning may require a senior citizen to attend meditation classes over a period of time.

Successful achievement of optimal best practice, $L_2$, is individualized, personal, and subjective. A person may use their level of actual best practice, $L_1$, as a point of reference in order to gauge into the setting of a personal best goal, known as 'goal of optimal best' (i.e., 'GOB') [17], which could assist in the achievement of $L_2$. A positive difference between $L_1$ and $L_2$ (i.e., successful progression from $L_1$ to $L_2$), denoted as '$+\Delta_{(L1-L2)}$', according to Phan and Ngu [17], is analogous to *a flow state* [13, 22, 23]. A flow state in a specific subject matter, also known as a 'state of flow', is intimately linked to the construction and setting of goals of optimal best for accomplishment, and the subsequent achievement of $L_2$. Moreover, this point of consideration regarding the notion of 'equivalency' (i.e., between $+\Delta_{(L1-L2)}$ and a state of flow) suggests that

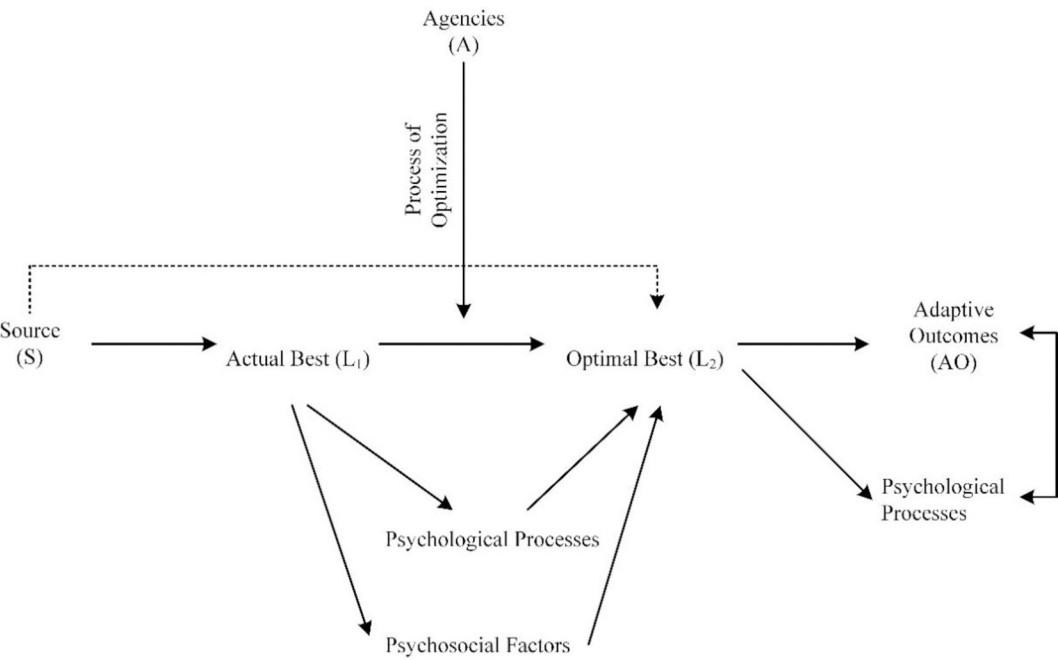

**Fig 1. Levels of best practice.** Source: [21].

optimization [1, 5, 18] could enact to facilitate and/or account for a person's state of flow. This conceptualization (i.e., $+\Delta_{(L1\ -L2)} \approx$ a flow state), likewise, posits that a person's sub-optimal experiences in a subject matter is closely aligned with their inactivity, malfunctioning, procrastination, etc.

## Conceptualization of a theoretical model for examination

Our focus of inquiry in the present study, quantitative in nature, closely aligns to the exploration of the underlying nature of both $L_1$ and $L_2$ [1, 5]. Our research interest in this matter specifically considers three major tenets and/or issues for examination: (i) what antecedents and/or sources of information would assist to account for a person's state of $L_1$, and to cause for the successful achievement of $L_2$?, (ii) what is the significance of both $L_1$ and $L_2$ in terms of explanatory and/or predictive power of future adaptive outcomes?–for example, does $L_2$ account for an improvement in academic engagement?, and (iii) what is the relationship between $L_1$ and $L_2$? At this stage, in terms of empirical research development, there are a number of cross-sectional and longitudinal studies that have yielded clear and consistent evidence, affirming the underlying nature of both $L_1$ and $L_2$.

A significant aspect of our research inquiry is closely aligned with a particular theoretical premise–namely, the encouragement, development, and cultivation of a 'motivational mindset' for personal growth. A motivational mindset, we propose, is associated with a person's *mental attitude* and *belief* towards learning, accomplishment, and flourishment. As such, a person's motivational mindset reflects her *optimistic* and *positive outlook* in life and about life. Feeling buoyant, energized, determined, and hopeful, from our point of view, would direct, compel, and govern a person to strive to achieve optimal best practice. For example, academically, having a motivational mindset would help to encourage, facilitate, and/or motivate a university student to strive to achieve optimal best experience in mathematics learning. In a similar vein, of course, it is plausible to contend that successful experience of optimal best

practice would highlight a person's motivational mindset. This postulation purports that one cannot achieve optimal best practice without a positive and motivational mindset. In other words, there is an intimate relationship between a motivational mindset and optimal best practice–that aside from external influences (e.g., assistance from a teacher in school), it is pertinent that a person adopts or has an appropriate mindset for learning, accomplishment, and flourishment. In their recent conceptual analysis article, Phan and his colleagues [8] proposed an interesting concept that we firmly believe may relate to the notion of a motivational mindset. According to the authors, in order to strive to achieve optimal best experiences in academic learning, say, a student has to consider 'personal best goals' for accomplishment. Personal best goals are goals that reflect a person's realistic aspiration and self-determination to achieve optimal best (e.g., My goal is to achieve much more for this subject than what I have indicated so far).

**Antecedents of adaptive outcomes.** The three aforementioned questions are interrelated with each other and, importantly, reflect the importance of the underlying nature of levels of best practice [17], as shown in Fig 1. Summarizing Fig 1, our focus of inquiry considers the following relationship:

$$\text{Antecedents} \rightarrow +\Delta_{(L1-L2)} \rightarrow \text{Adaptive Outcome}$$

From the above summary, our examination first of all focuses on the impact of and antecedent or source of information on the two levels of best practice and a related adaptive outcome, namely, academic engagement. We consider the potential influences of three contrasting psychological variables on both $L_1$ and $L_2$. An examination of the educational psychology literature indicates that there are numerous psychological variables (e.g., self-efficacy for learning) [24–26] and psychosocial factors (e.g., the impact of the home environment) [27, 28], which could function as antecedents of future adaptive outcomes [29–32]. By all account, choosing which psychological variable and/or psychosocial factor to examine is a personal feat, coinciding with existing research development, theoretical premise(s), a researcher's interest, etc. Our choosing of three psychological variables (i.e., self-concept, optimism, and pessimism) for undertaking, in this case, is based on three important factors: (i) clear and consistent evidence from existing research development, which indicates the predictive role of the psychological variable (e.g., the predictive effect of optimism on academic performance) [33], (ii) advancing knowledge and theoretical understanding of the theoretical premise of the process of human optimization [1, 5, 18], which contends that different types of psychological variables may help to optimize a person's state of functioning, and (iii) intellectual curiosity to seek theoretical understanding into the comparative antecedent roles of the chosen psychological variables (e.g., optimism *versus* pessimism).

As detailed in this section of the article, we purposely chose psychological variables that would cast contrasting effects (i.e., positive effect *versus* negative effect) on levels of best practice and/or different types of adaptive outcomes (e.g., academic performance)–

i. *Academic self-concept*, which is defined as "a person's self-evaluations of their academic capability" [34], has been noted to positively associate with academic performance and other related adaptive outcomes [e.g., 35–37]. For example, in a longitudinal study that used autoregressive structural modelling, Green and her colleagues [34] found that academic self-concept positively influenced the concept of 'positive attitudes toward school' across Time 1 (e.g., β = .28, $p < .001$) and Time 2 (e.g., β = .22, $p < .001$). In a similar vein, one of the earlier prominent studies using longitudinal data [38] found evidence of reciprocal effects between self-esteem, academic self-concept, and academic achievement (e.g., Time 1 academic self-concept → Time 2 academic achievement → Time 3 academic self-concept

*versus* Time 1 academic achievement → Time 2 academic self-concept → Time 3 academic achievement).

ii. *Optimism*, which is defined as "an expectation that people will endure positive experiences" [39], is a positive psychological concept that differs from 'pessimism', which is negative and may yield detrimental consequences [40, 41]. Optimism, often aligns to the paradigm of positive psychology [9, 10], is a potent predictor of different types of adaptive outcomes [e.g., 33, 42–44]. In an earlier study, for example, Tschannen-Moran and her colleagues [45] found that optimism positively influenced academic achievement ($\beta$ = .73, $p < .001$). Non-academically, a recent study [43] found that optimism exerted a positive effect on perceived work satisfaction ($\beta$ = .29, $p < .01$).

iii. *Pessimism* is somewhat different from academic self-concept and optimism for its negative nature [40, 41]. Pessimism, according to the American Psychological Association, is defined as the "attitude that things will go wrong and that people's wishes or aims are unlikely to be fulfilled" (Source: https://www.verywellmind.com/is-it-safer-to-be-a-pessimist-3144874#citation-1). Unlike optimism, pessimism is opposite in nature and in this case, closely aligns with maladaptive functioning and negative outcomes (e.g., low academic performance) [e.g., 46–48]. For example, a recent study that involved the German higher education context [46] found that pessimism negatively influenced needs for autonomy ($\beta$ = -.25, $p < .05$), needs for competence ($\beta$ = -.24, $p < .05$), needs for relatedness ($\beta$ = -.30, $p < .01$), and satisfaction with life ($\beta$ = -.34, $p < .001$).

From the preceding sections, we postulate that both academic self-concept and optimism would positively account for a person's testament of $L_1$ (e.g., academic self-concept → $L_1$, +$\beta$ value) and/or $L_2$ (e.g., optimism → $L_2$, +$\beta$ value) and that, in contrast, pessimism would negatively influence the two levels of best practice (e.g., pessimism → $L_1$, -$\beta$ value). In this analysis, consistent with previous inquiries [1, 5], we contend that academic self-concept and/or optimism would act as sources of motivation and optimization, helping to facilitate a person to strive for best practice in a subject matter. In a similar vein, of course, we posit that academic self-concept, optimism, and pessimism would cast contrast influences on another achievement-related outcome–namely, *academic engagement* for effective learning [46, 49–51]. For example, in this case, we speculate that pessimism would negatively influence academic engagement for learning, whereas both academic self-concept and optimization would exert positive influences.

We acknowledge, of course, that other psychological, psychosocial, and educational variables could also operate as antecedents and/or optimizing sources of both $L_1$ and $L_2$. In terms of explanatory account, as we have detailed, two contrasting influences of psychological, psychosocial, and/or educational variables may arise: positive influence (e.g., optimism → $L_2$, +$\beta$ value) *versus* negative influence (e.g., pessimism → $L_2$, -$\beta$ value). Phan and his colleagues, for example, have focused predominantly on positive influences on both $L_1$ and $L_2$. For example, the authors found some interesting evidence pertaining to the antecedent → $L_1$ relationship and the antecedent → $L_2$ relationship–(i) from Phan, Ngu [52], the effect of self-efficacy for academic learning on $L_1$ ($\beta$ = .43, $p < .001$) and the effects of energy ($\beta$ = .24, $p < .01$), $L_1$ ($\beta$ = .28, $p < .001$), personal resolve ($\beta$ = .41, $p < .001$), and sustaining ($\beta$ = .14, $p < .01$) on $L_2$, and (ii) from Phan and Ngu [21], the effect of academic striving ($\beta$ = .21, $p < .01$) on $L_1$ and the effects of $L_1$ ($\beta$ = .33, $p < .001$) and personal resolve ($\beta$ = .17, $p < .01$) on $L_2$.

**Predictive effects of best practice.** In terms of the underlying nature of best practice [1, 5], we expect to find that both $L_1$ and $L_2$ would account for and/or explain achievement of other adaptive outcomes. Optimal best practice, $L_2$, in a specific subject matter is positive and

motivational, reflecting an internal state of flow, optimism, and proactivity. As such, successful accomplishment of $L_2$ would function as a source of information, motivating and assisting a person to undertake and achieve other adaptive outcomes. For example, we speculate that in this case, a university student's successful achievement of $L_2$ in a subject matter would motivate and assist her to proactively engage in other subject matters (i.e., $L_2 \rightarrow$ engagement, $+\beta$ value). Existing research development, interestingly, has produced clear and consistent evidence, highlighting the predictive and explanatory power of $L_2$ –for example, in an earlier study, Phan and his colleagues [53] noted that $L_2$ exerted a positive effect on personal well-being ($\beta =$ .28, $p < .001$). This evidence, albeit preliminary at present, seems to suggest that a state of $L_2$ in a subject matter is not final and may instead function as a central variable or mechanism, which then could associate with and/or predict other types of adaptive outcomes.

In the context of the present study, does a state of $L_1$ and/or $L_2$ predict academic engagement in a subject matter (i.e., $L_1 \rightarrow$ academic engagement *versus* $L_2 \rightarrow$ academic engagement)? This reflective question for examination contends that a state of $L_1$ and/or $L_2$ could act as a central source of motivation, helping to facilitate the improvement of other adaptive outcomes. Existing research inquiries, interestingly, have noted the direct effects of other psychosocial and psychological variables on academic engagement–for example: teacher support [54], peer social relationship [55], goal progress [56], subjective task value [50], and optimism [46]. As such, testament of the positive predictive effect of a state of $L_1$ and/or $L_2$ on academic engagement would add value and support the proposed underlying nature of best practice [1, 2] for further development. For example, unclear at the present time, is whether there is a potential psychological mechanism and/or sub-mechanisms, which could account for and explain the predictive power of a state of $L_1$ and/or $L_2$. What is it within a state of $L_2$, for example, that would help facilitate the successful achievement of academic engagement?

## Significance of the present study

The present study, situated to the learning context of Spain, is innovative for its attempt to elucidate theoretical understanding of the recently developed inquiry–namely, the *contextualization and importance of best practice* [1–3]. This research inquiry is sound and logical, providing evidence that could serve to inform, suggest, and/or recommend a number of educational and/or non-educational benefits for consideration. For example, within the context of academic learning, how could an educator use a student's successful achievement of $L_2$ to motivate her to strive for success in other domains of functioning? In a similar vein, non-academically, how could a health worker assist a senior citizen to strive to achieve optimal health during the Covid-19 pandemic period? In terms of sports, differently, how could a coach help his team achieve optimal best and succeed in the forthcoming season? These real-life questions, we contend, emphasize the importance of the considered viewpoint of best practice as a distinct theoretical concept or orientation that could contribute to our understanding of human agency [57, 58].

In sum, the present study attempts to explore and test the proposed theoretical model that is shown in Fig 2, detailing the central role of best practice. There are three major interrelated inquiries for examination, namely: (i) the direct predictive effects of three comparable and comparative antecedents on the two levels of best practice (e.g., optimism $\rightarrow L_1$ *versus* academic self-concept $\rightarrow L_1$) and academic engagement (e.g., pessimism $\rightarrow$ academic engagement *versus* optimism $\rightarrow$ academic engagement), (ii) the predictive effects of the two levels of best practice on academic engagement (e.g., $L_1 \rightarrow$ academic engagement *versus* $L_2 \rightarrow$ academic engagement), and (iii) the potential mediating mechanisms of the two levels of best practice between the three mentioned antecedents and academic engagement (e.g., academic self-

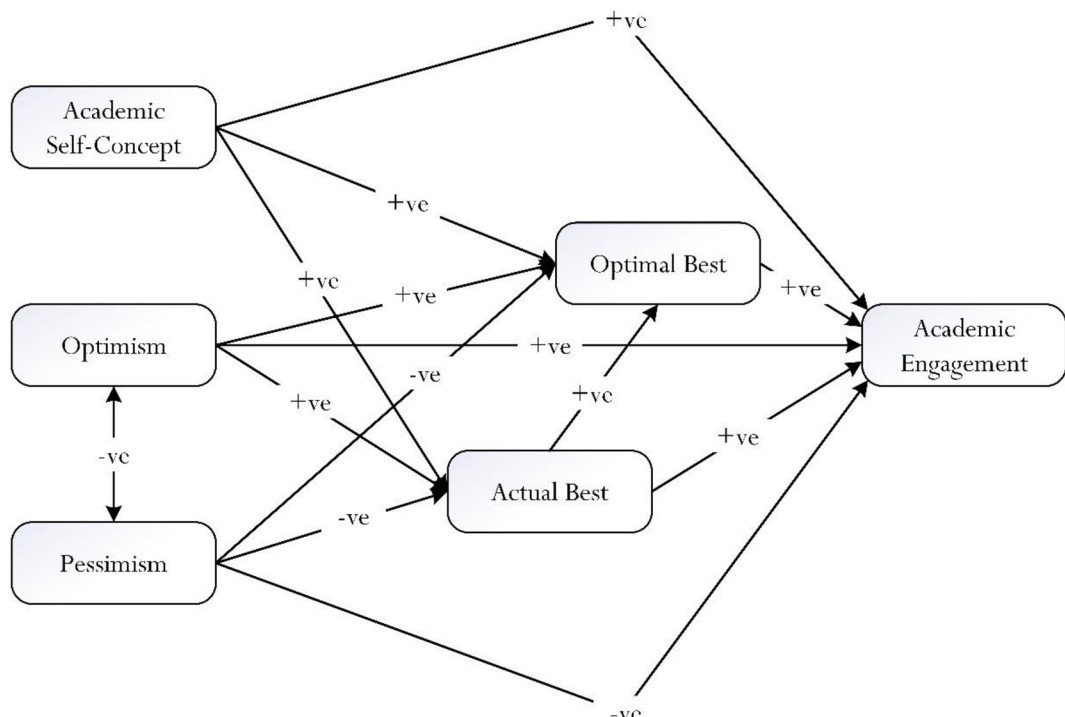

**Fig 2. Conceptualization of study for investigation.** Note: +ve = positive predictive effect, -ve = negative predictive effect.

concept → $L_1$ (i.e., potential mediator) → academic engagement *versus* academic self-concept → $L_2$ (i.e., potential mediator) → academic engagement). The three research inquiries, overall, may yield fruitful theoretical insights for the purpose of applied practice, academically and/or non-academically. For example, evidence of the relationship between academic self-concept, a state of $L_2$, and academic engagement (i.e., academic self-concept → $L_2$ → academic engagement, +β value) could inform us of the potentiality to design and/or to structure an in-class intervention for usage, which closely aligns with academic self-concept. In a similar vein, it is plausible for us to consider a state of $L_2$ as a point of reference for development and/or implementation.

As an important side note, the present study may also offer relevant and insightful information that could lend support to our discussion of what we term as a 'motivational mindset'. With reference to our earlier mentioning, successful experience of optimal best practice could potentially reflect and espouse a person's positive and motivational mindset. Evidence of optimal best practice, in this case, may entail and/or consist of different types of functioning. As the present study conceptualizes, one particular type of functioning that may affirm the achievement of optimal best practice is its explanatory and predictive effect on academic engagement. Such considered viewpoint (i.e., a motivational mindset would assist to facilitate the achievement of optimal best practice), we contend, is significant in terms of theoretical contributions to the study of optimal best practice [1–3].

## Method

### Design

The research design was observational, descriptive, cross-sectional and non-randomized. The participants came from several Andalusian universities. The data were collected at the end of the 2020/2021 academic year. The inclusion criteria were: (i) to be a student of the Degree and

Master's Degree in Secondary Education Teaching (presential modality); the exclusion criteria were: (i) not consenting to the use of the study data; (ii) not completely filling in the data collection form.

## Sample size and participants

An a priori analysis was carried out on the statistical power of the adequate sample size to meet the study objective. Using the *Free Statistics Calculator* v.4.0 software (50), it was estimated that a minimum of 661 participants were needed for $f^2$ = .19 effect sizes with a statistical power level of .95 and a significance level of $\alpha$ = .05 in a structural equation model with six latent variables and 39 observable variables. In this study, 681 physical education pre-service teacher students (395 women; 285 men; 1 other) of the Degree (61.4%) and Master's Degree in Secondary Education Teaching (38.6%) from four public Andalusian universities (Spain) participated. The age of the participants ranged between 18 and 57 years ($M$ = 23.85; $SD$ = 5.35). There were no missing values in the included sample data.

## Instruments

**Current best practice.** The shorter version of the subscale of the Optimal Outcomes Questionnaire [2], called Current Best Practice, was adapted. The shorter version is composed of 5 items to assess the perception of a person's actual competence at present. The responses were collected on a Likert scale with a range of responses between 1 (strongly disagree) and 5 (strongly agree). As this is the first time that the scale has been used in the Spanish context, the factorial structure was first evaluated by CFA (*Confirmatory Factor Analysis*), which showed adequate goodness-of-fit: $\chi^2$/df (chi square/degrees of freedom) = 2.838, $p$ = .023; CFI (*Comparative Fit Index*) = .982; TLI (*Tucker–Lewis Index*) = .956; RMSEA (*Root Mean Square Error of Approximation*) = .069 [90% Confidence Interval (CI) = .037; .082]; SRMR (*Standardized Root Mean Square Residual*) = .034. The internal consistency values were acceptable: composite reliability (CR) = .71, Cronbach's alpha ($\alpha$) = .74, index $H$ = .74.

**Optimal best practice.** The shorter version of the subscale of the Optimal Outcomes Questionnaire [2], called Optimal Best Practice, was adapted. This version is composed of 5 items to assess the perception of a person's competence maximization at present. The responses were collected on a Likert scale with a range of responses between 1 (strongly disagree) and 5 (strongly agree). As this is the first time that the scale has been used in the Spanish context, the factorial structure was first evaluated by CFA (*Confirmatory Factor Analysis*), which showed adequate goodness-of-fit indices: $\chi^2$/df (chi square/degrees of freedom) = 1.188, $p$ = .305; CFI (*Comparative Fit Index*) = .997; TLI (*Tucker–Lewis Index*) = .992; RMSEA (*Root Mean Square Error of Approximation*) = .026 [90% Confidence Interval (CI) = .000; .126]; SRMR (*Standardized Root Mean Square Residual*) = .023. The internal consistency values were acceptable: CR = .73, Cronbach's alpha ($\alpha$) = .71, $H$ index = .71.

**Academic self-concept.** The Spanish adaptation to the university context [59] of the Academic Self-Concept Scale by Matovu [60] was used. The scale consists of six items that measure academic confidence (3 items, e.g., "I am able to help my course mates in their schoolwork") and academic effort (3 items, e.g., "I pay attention to the lecturers during lectures."). The responses were collected on a Likert scale with a range of responses between 1 (strongly disagree) and 7 (strongly agree). Academic self-concept was calculated as the average value of the scores for the two factors comprising it. The goodness-of-fit indices of this scale were acceptable: $\chi^2$/df = 4.391, $p$ < .0001; CFI = .975; TLI = .954; RMSEA = .071 (90%CI = .048, .075), SRMR = .030. The internal consistency values were acceptable: academic self-confidence: CR = .72, $\alpha$ = .71, $H$ index = .73; academic effort: CR = .78, $\alpha$ = .77, $H$ index = .80.

**Life Orientation Test-Revised.**   The Spanish version of the Life Orientation Test-Revised (LOT-R) [61] was used. This instrument is a test to measure individual differences in generalized optimism versus pessimism. The scale consists of ten items and two factors: optimism (3 items, e.g., In uncertain times, I usually expect the best) and pessimism (3 items, e.g., I hardly ever expect things to go my way); moreover, four items were included to disguise (to some extent) the underlying purpose of the test. The responses were collected on a Likert scale with a range of responses between 0 (strongly disagree) and 4 (strongly agree). To obtain the total score, the items referring to pessimism were inverted. The goodness-of-fit indices of this scale were acceptable: $\chi^2/df$ = 2.144, $p$ = .036; CFI = .991; TLI = .980; RMSEA = .041 (90% CI = .010, .070), SRMR = .023. The internal consistency values were acceptable: optimism: CR = .70, $\alpha$ = .71, $H$ index = .72; pessimism: CR = .72, $\alpha$ = .68, $H$ index = .75. Although the pessimism subscale showed an $\alpha < .70$, it can still be considered marginally acceptable (54) given the small number of factor items.

**Academic engagement.**   The student version of the *Utrecht Work Engagement Student Scale* (UWES-SS) [62] was used. The scale is composed of 17 items that are arranged into three factors: vigour (6 items) (e.g., "I can continue working for very long periods at a time"), dedication (5 items) (e.g., "I am proud on the work that I do"), absorption (6 items) (e.g., "I feel happy when I am working intensely"). The responses were collected on a Likert scale with a range of responses between 1 (strongly disagree) and 5 (strongly agree). Academic engagement was calculated as the average value of the scores for each of the factors that comprise it. The goodness-of-fit indices of this scale were acceptable: $\chi^2/gl$ = 4.689, $p < .0001$; CFI = .955; TLI = .929; RMSEA = .073 (90%CI = .060, .087), SRMR = .037. The internal consistency values were acceptable: vigour: CR = .71, $\alpha$ = .71, $H$ index = .71; dedication: CR = .77, $\alpha$ = .73, $H$ index = .75; absorption: CR = .73, $\alpha$ = .73, $H$ index = .74.

## Procedure

First, the Current Best Practice and the Optimal Best Practice were adapted, following the indications of Muñiz et al. [63] in relation to reverse translation (*back-translation)* and retro-translation. Next, those responsible for the Master's Degree in Secondary and Upper-secondary Education Teaching, Vocational Training and Language Teaching were contacted to request their permission and ask for their collaboration in the research. The students were then contacted by email. Data were collected using an online form throughout May 2021. The form briefly explained the importance of the research, the anonymity of the responses, the way to complete the scale, that the responses given would not affect any qualification in any way, and that the participants could stop participating in the study at any time. All participants gave their written consent to be included in the study. The research was conducted in accordance with the Declaration of Helsinki and the protocol was approved by the Bioethics Committee of the University of Almería (Spain) (Ref: UALBIO2021/009).

## Risk of bias

There was no sample randomization since the sampling was for convenience, although there was blinding between the participants and the researchers in charge of the data treatment and analysis. With respect to selection bias, participation was voluntary and communication with participants was conducted by email.

## Data analysis

First, the descriptive statistics of each item from the Current Best Practice and the Optimal Best Practice were calculated and the factorial structure of each scale was evaluated with CFA.

Standardized regression weights were considered minimally acceptable with values $\geq$.32 (57). The descriptive and correlation analyses were calculated with SPSS 27 (IBM, Chicago, IL, USA). The factorial model was assessed separately for each instrument by CFA using AMOS 26. The evaluation of the models was based on the following goodness-of-fit indices: values of the $\chi^2$/df ratio, CFI, TLI, RMSEA with its 90% confidence interval (CI), and SRMR. For the $\chi^2$/df ratio, values <2.0 or <5.0 are considered excellent [64] or acceptable [65], respectively, < .95 or between .90 and .95 (CFI and TLI), < .06 or .10 (RMSEA) indicate an excellent or marginally acceptable fit, respectively, and values < .08 for SRMR. Construct reliability was evaluated using the $H$ coefficient, with values $\geq$.70 being considered acceptable, and recommended for reliability when using an SEM approach (60). Cronbach's alpha and composite reliability were also calculated, with values $\geq$.70 being acceptable [66]. The predictive relationships of the structural equation model (SEM) were verified using a path analysis model performed using AMOS 26. Following Kline [67], this technique was chosen to ensure the reliability of the results, because analysis using latent variables would have involved the presence of a low proportion of cases / free parameters. The model evaluation was carried out taking into account the goodness-of-fit indices indicated above (i.e., $\chi^2$/df, CFI, TLI, RMSEA with 90% CI, and SRMR).

In the hypothesized model, the following direct relationships were established: academic self-concept, optimism and pessimism with current best practice, optimal best practice, and academic engagement; current best practice with optimal best practice and academic engagement; optimal best practice with academic engagement. Indirect relationships were established between academic self-concept, optimism and pessimism and academic engagement through current best practice and optimal best practice, as well as between current best practice and academic engagement through optimal best practice. Taking into account the Mardia coefficient values (1.99; $p < .05$), analyses were performed using the maximum likelihood method and the 5000-iteration *bootstrapping* procedure [67]. In addition, $R^2$ was used for the effect sizes (ES) to improve the results interpretation, since it estimates the degree of influence by quantifying the variance percentage of the dependent variable explained by the predictors [68]. The cut-off points were: 02, .13, and .26, for small, medium, and large effect sizes, respectively [69]. Furthermore, the confidence intervals (CI95%) were calculated to ensure that no $R^2$ value was < .02, as this is the minimum required for its interpretation.

# Results

## Preliminary results

Table 1 presents the descriptive statistics and correlations between the variables included in the study.

**Table 1. Descriptive statistics and correlation between variables.**

| Variable | Range | *M* | *SD* | Q1 | Q2 | 2 | 3 | 4 | 5 | 6 |
|---|---|---|---|---|---|---|---|---|---|---|
| 1.Academic Self-Concept | 1–7 | 5.32 | .97 | -.31 | -.53 | .32** | -.14** | .51** | .41** | .55** |
| 2.Optimism | 0–4 | 2.64 | .85 | -.34 | -.15 | | -.40** | .35** | .28** | .31** |
| 3.Pesismism | 0–4 | 1.76 | .89 | .08 | -.33 | | | -.20** | -.07 | -.05 |
| 4.Current Best | 1–5 | 3.93 | .71 | -.38 | -.45 | | | | .47** | .45** |
| 5.Optimal Best | 1–5 | 3.70 | .59 | -.09 | -.13 | | | | | .37** |
| 6.Academic Engagement | 1–5 | 3.48 | .80 | -.31 | -.09 | | | | | |

Note.

**The correlation is significant at the .01 level; *M* = Mean; *SD* = Standard deviation; Q1 = Skewness; Q2 = Kurtosis.

## Main results

The predictive SEM model showed the following acceptable goodness-of-fit indices: $\chi^2/df = 2.977$, $p = .084$; CFI = .998; TLI = .957; RMSEA = .054 (90%CI = .000; .083; $p_{close} = .338$), SRMR = .010. The explained variance was 28% for optimal best practice, 30% for current best practice, and 37% for academic engagement. After controlling for gender, the direct relationship between pessimism and current best practice was not statistically significant in the SEM model. The following direct effects were statistically significant: between academic self-concept and current best, and optimal best and academic engagement; between optimism and current best, optimal best and academic engagement; between pessimism and current best, optimal best and academic engagement; between current best and optimal best and academic engagement; and between optimal best and academic engagement. Overall, the final solution in Fig 3.

Regarding the mediating variables, current best was a mediator between academic self-concept and academic engagement ($\beta = .08$; $p < .001$), and between optimism and academic engagement ($\beta = .04$; $p < .001$). Optimal best was a mediator between academic self-concept and academic engagement ($\beta = .04$; $p < .05$), optimism and academic engagement ($\beta = .01$; $p < .001$), pessimism and academic engagement ($\beta = -.01$; $p < .05$), and current best and academic engagement ($\beta = .04$; $p < .05$). The total effects of academic self-concept on academic engagement mediated by current best were $\beta = .47$ ($p < .001$); when mediated by optimal best, they were $\beta = .41$ ($p < .05$). In addition, the multiple mediation (i.e., current best and optimal best) between academic self-concept and academic engagement was $\beta = .13$ ($p < .001$), and the total effects were $\beta = .52$ ($p < .001$). The total effects of optimism on academic engagement mediated by current best were $\beta = .17$ ($p < .001$); when mediated by optimal best, they were $\beta = .14$ ($p < .05$). The multiple mediation (i.e., current best and optimal best) between optimism and academic engagement was $\beta = .06$ ($p < .001$), and the total effects were $\beta = .19$ ($p < .001$).

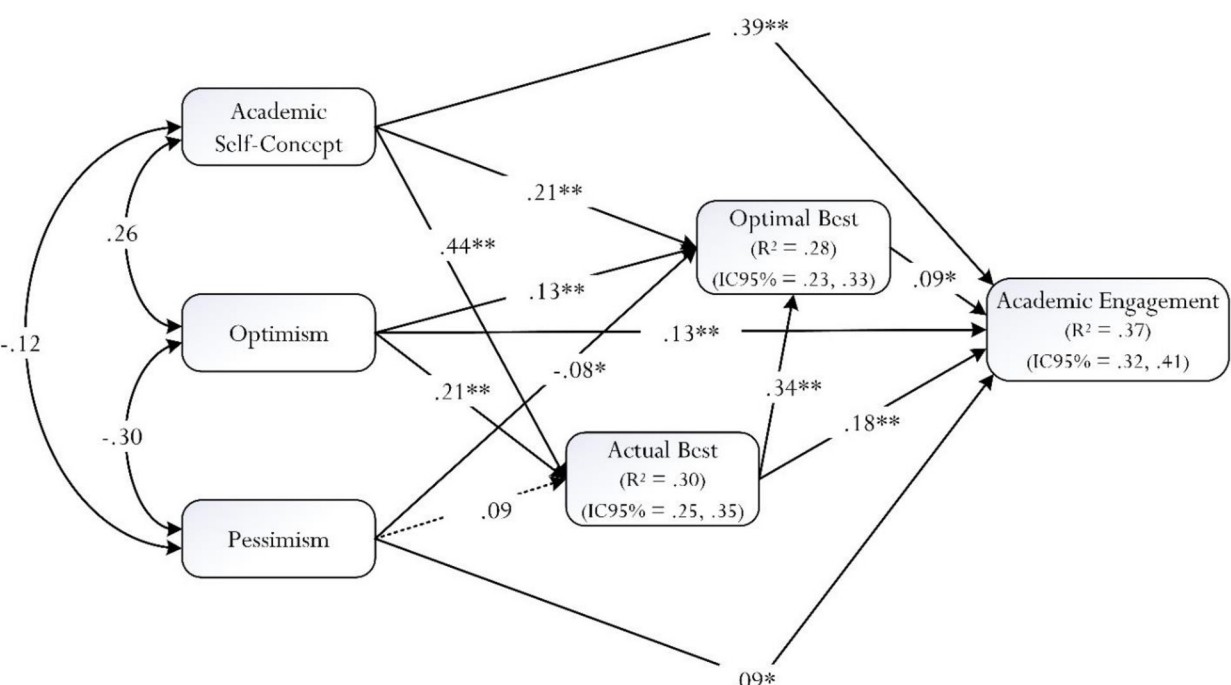

**Fig 3. Predictive relationships for consideration.** Note: $^{**}p < .001$; $^{*}p < .05$. $R^2$ = Explained variance; CI = Confidence interval. The dashed lines represent non-significant relationships.

The total effects of pessimism on academic engagement mediated by optimal best were $\beta = .08$ ($p < .05$). The total effects of current best on academic engagement mediated by optimal best were $\beta = .21$ ($p < .05$). Finally, Fig 1 shows the CI(95%) of $R^2$, which can be considered measures of ES [68] and, in all cases, these are large.

## Discussion

The present study is significant for its inquiry into the operational functioning of optimal best practice [1–3]. Optimal best practice is a positive concept that may coincide with a flow state [11–13] and a person's experience of flourishing [14–16]. Furthermore, from our conceptualization, evidence of optimal best practice in a subject matter (e.g., optimal learning experience in mathematics) may indicate person's state of motivation to strive to achieve success in other related subject matters. Our research interest has led us to propose a theoretical-conceptual model for examination, which involved two interrelated questions: (i) what sources of information or antecedents may serve to account for a person's achievement of optimal best, $L_2$, and (ii) does a person's successful achievement of optimal best, $L_2$, result in his proactive engagement for learning? In total, as shown in Fig 3, evidence ascertained from SPSS AMOS 26 indicates strong empirical support for our hypothesized *a priori* model. Importantly, likewise, the solution shown in Fig 3 may offer innovative methodological and teaching insights for consideration.

### Theoretical contributions

The underlying nature of optimal best practice [1–3], as shown in Figs 1 and 2, consists of three major elements: (i) antecedents that may positively predict optimal best practice, (ii) the process of optimization that may help to optimize optimal best practice, and (iii) the predictive and explanatory effect of optimal best practice. Over the past several years, we have undertaken a few notable studies, which have resulted in interesting findings for the purpose of cross-cultural comparison. Limited by our use of non-experimental data, we were not able to delve into the intricacy of the 'optimization' of optimal best practice [18]. At best, our research inquiries were more in line with a focus on associative patterns, or explanatory and predictive relationships, between antecedents and consequences of optimal best practice.

**Antecedents of optimal best practice.** Our *a priori* model for examination, as shown in Fig 2, is unique for its considered viewpoint and proposition, which could help explain the underlying nature of optimal best practice [1–3]. Evidence obtained via means of SEM [70, 71] indicates different types of antecedents may account for a person's achievement of optimal best in a specific subject matter. As Fig 3 illustrates, there are two contrasting possibilities: (i) perceived 'positive' antecedents (e.g., optimism → $L_2$, +ve effect), which we want to encourage, promote, and/or cultivate *versus* (ii) perceived 'negative' antecedents (e.g., pessimism → $L_2$, -ve effect), which we want to negate, discourage, and/or dissuade. Optimal best practice, as a recap, emphasizes a person's internal state of 'maximization' of functioning (e.g., the maximization in cognitive functioning in mathematics learning). As such, on a daily basis, we want to assist and to motivate individuals to strive for optimal best practice in different domains of functioning. A football coach, for instance, may wish to consider different means, which in turn could help his players to strive and achieve optimal best in the forthcoming season. Why is this the case? Summed up briefly, such successful of optimal best practice experience (e.g., successful accomplishment of scoring 20 goals in the 2022/2023 season) would espouse and reflect a person's positive outlook about life, a subject matter, a personal situation, etc.

Distinction in characteristics of antecedents of optimal best practice (e.g., optimism *versus* pessimism) may require further examination. Interestingly, for example, we note that despite its negative characteristics [72–74], pessimism exerts both a positive effect on academic

engagement and a negative effect on optimal best practice. This finding is somewhat perplexing (i.e., the positive effect of pessimism on academic engagement), given that both optimal best practice and academic engagement are analogous with each other (i.e., both constructs are positive in nature). Moreover, pessimism is negative and closely aligns with a state of demotivation and a perceived sense of helplessness [75–77], resulting in a person giving up and/or avoiding work altogether. As such, unlike that of a state of optimism, we find it somewhat uncharacteristic for a state of pessimism to positively associate with academic engagement. This noting is poignant, especially when we compare the small negative effect of pessimism with the positive effect of optimism on optimal best practice.

**Consequence of optimal best practice.**   Similar to previous research studies [e.g., 53, 78], our regression analysis showed that optimal best practice served as a positive predictor of academic engagement. As Fig 3 indicates, successful experience of optimal best practice helps a student to proactively engage in academic learning. This evidence highlights the potentiality for optimal best practice to act as a source of motivation, helping to facilitate, guide, and/or motivate person's intent to engage in a specific course of action, which could potentially result in successful accomplishment. Such discourse, likewise, would consider an opposite pattern– that unsuccessful experience of optimal best practice or, alternatively, a low level of optimal best practice would closely align with a state of disengagement [79–81]. Moreover, evidence as shown in Fig 3 suggests that optimal best practice in itself is not an end-all outcome for accomplishment but that, rather, it acts as a central variable, which may help to mediate other effects. For example, as reported earlier on, optimal best practice may mediate the effects of academic self-concept, optimism, pessimism, and current best practice on academic engagement (e.g., academic self-concept → optimal best practice → academic engagement, where optimal best practice serves as a mediator).

It is a valuable feat for us to affirm and, by the same token, to facilitate and strengthen the potent role of optimal best practice. That optimal best practice could and/or would explain and positively predict, as shown in Fig 1, is a significant feat, highlighting its unique characteristics [1–3], which one may capitalize on. One notable characteristic of optimal best practice, in this case, relates to its upbeat and optimistic state, serving to optimize and motivate a person's specific functioning (e.g., cognitive functioning) for a particular context. Indeed, existing discussions and research inquiries have likened the nature of optimal best practice to the paradigm of positive psychology [9, 10], which seeks to explore pathways, means, and/or opportunities that could strengthen a person's experience of flourishing.

Successful experience of optimal best practice in a subject matter [1–3] in itself is a distinction for formal recognition. Moreover, however, a person's indication of her successful achievement of optimal best practice may serve to highlight the positivity of an organization's, community's, company's, or institution's social milieu. That the creation of a social milieu, or social environment, that is conducive and motivational for optimal best experiences. For example, non-academically, a training facility may attribute and assist a professional football player's optimal best in scoring for his football club. In a similar vein, academically, a school's particular ethos may motivate students to strive to achieve optimal academic learning experiences. In their recent conceptual analysis article, Phan and his colleagues [8] proposed an interesting conceptualization–that achieving optimal best from actual best practice in a subject matter (Note: $\beta = .34$, $p < .01$ in the present study) would *equate* to a state of flow [22, 82, 83]. In other words, according to the authors, a positive difference between actual best practice and optimal best practice, i.e., $+\Delta_{(L2 - L1)}$, is analogous to a state of flow (i.e., $+\Delta_{(L2 - L1)} \approx$ a flow state). This considered viewpoint, we contend, is insightful as it espouses one notable aspect for consideration: that achieving optimal best practice significant and advantageous, as this successful experience would also initiate and sustain a state of flow.

## Practical and methodological implications for consideration

The present study has also provided detailed practical and methodological implications for us to consider. Foremost from our conceptualization, based on existing theorizations [1–3] and empirical research inquiries [e.g., 21, 53], is the use of a 'positive and motivational mindset' to flourish and achieve. A student's positive and motivational mindset, we contend, espouses his self-belief and personal conviction that he could achieve anything that is within the realm of actuality. Moreover, from our point of view, a positive and motivational mindset would indicate that achievement of optimal best practice is not resolute (i.e., can a person achieve optimal best practice in a subject matter?–yes or no?), and that its possibility is dependent on different psychological, educational, and psychosocial factors. Academically, for example, what would educators have to do in order to encourage and promote optimal best learning experiences? Optimal best practice in academic learning, in this sense, is a priority and, more importantly, may indicate the cornerstone of successful schooling of an institution.

**Educational implications for consideration.**   As conceptualized in Fig 2, optimal best practice may operate as a central variable, which in turn would mobilize, govern, and/or direct a person's thought and behavioural patterns for effective and/or non-learning experiences. With this in mind, it is plausible for us to consider the following for educational (or non-educational) purposes:

i. Optimal best practice as an important source of information, which could operate to optimize and motivate a person's thought and behavioural patterns [1–3]. This considered viewpoint contends that perhaps, one (e.g., a teacher) could encourage individuals to focus on striving to achieve optimal best practice. Optimal best practice is personal and subject to individual variations between individuals, reflecting their differing levels of optimism, motivation, aspiration, positive outlook, etc. Striving to achieve optimal best practice, which is a personal best goal for accomplishment [8], may help motivate individuals to engage in the learning and/or non-learning processes. Within the context of academic learning in a specific subject or a variety of subjects, it is plausible for educators to use a psychological concept known as personal resolve [5, 6, 21], which we cited earlier, to encourage, promote, and/or facilitate optimal best practice. Personal resolve, in brief, is defined as a person's "internal state of decisiveness and resolute to strive for optimal achievement best in an optimistic manner" [84]. As such, from this definition, personal resolve emphasizes the importance of a student's self-determination to overcome any obstacles that may arise, guiding and directing her to engage in a purposive act to achieve optimal best (i.e., personal resolve → $L_2$) [21, 84]. In this sense, personal resolve is analogous to an internal state of mental fortitude, serving as a potent source of motivation for learning and non-learning purposes. Timely use of positive feedback in class, for example, may instil and/or encourage a heightened state of personal resolve, which in turn could motivate a student to strive for optimal best practice.

ii. Design pathways, means, and/or opportunities, which could stimulate motivation for achievement of optimal best practice. Our present findings, for example, indicate support for the potential use of academic self-concept [e.g., 35–37], optimism [e.g., 33, 42–44], and other positive constructs (e.g., hope) [85–87] that could, indeed, serve to encourage and/or to stimulate the striving of optimal best. For example, positive verbal discourse (e.g., encouraging feedback, which may reflect a state of hope and inspiration) may instil a level of academic self-concept for learning, in turn helping to motivate students to strive for optimal best experiences. In a similar vein, encouraging a student to show optimism regardless of perceived difficulties, hardships, etc. could also help.

Interesting subject contents and/or contextual matters that have daily relevance, likewise, may assist to encourage individuals to strive to achieve optimal best practice. In terms of academic learning, for example, interesting subject contents (e.g., the study of paranormal psychology) [88] may encourage situational interest, intellectual curiosity, and aspiration for mastery, all of which could motivate a student to strive to achieve optimal best experiences. Difficult and/or uninterest subject contents (e.g., the learning of algebra) [20] that are perceived as being 'boring' [89], in contrast, are more likely to demotivate, weakening one's interest, intellectual curiosity, aspiration to strive to achieve mastery. In a similar vein, a contextual matter or situation that has life relevance and/or daily applicability (e.g., the practice of meditation) [90–92] is more likely to enhance, stimulate, and/or facilitate, motivating and guiding a person to strive for optimal life functioning.

**Methodological insights for development.** By all account, the present study has also provided a few interesting methodological insights for consideration. Foremost as a major limitation is our use of cross-sectional, non-experimental data to explore the antecedents and consequences of optimal best practice. This caveat, from our point of view, underlines more than just our inability to gauge into longitudinal associate patterns in relationship for further development [93–95]. Rather, as existing theorizations have shown [1, 8, 18], achieving optimal best practice is in itself a longitudinal or a developmental process. In other words, as indicated earlier on, the relationship between actual best practice and optimal best practice (i.e., $+\Delta_{(L2 - L1)} \approx$ a flow state) is non-instantaneous. As such, we contend that accurate validation of the $L_1$-$L_2$ relationship and its associations with other psychological processes [1, 3, 53] would require the use of longitudinal data–for example, optimism at Time 1 could exert a temporally-displaced effect on actual best practice at Time 2, which in turn would account for achievement of optimal best practice at Time 3, etc. This postulation for the use of longitudinal data, importantly, contends that achieving optimal best practice is progressive and requires personal development, reflection, etc.

Relating to the importance of 'methodological appropriateness' [18], we recently conceptualized a proposition for further development: the possibility that an 'optimal best practice window' or, alternatively, a 'window for optimal best practice' could exist, helping us to identify a critical period for optimal growth. This conceptualization is based on philosophical reasoning and has no empirical evidence, at present, for substantiation. Our conceptualization of the possibility of an optimal best practice window is shown in Fig 4. For example, as Fig 4 depicts, which period in a person's lifespan is most critical in the maximization of her optimal best practice for a particular context (e.g., academic learning)? Is it logical to postulate that, perhaps, there is a specific period in life, which one could flourish and maximize her state of functioning? To consider this possibility, we could take one of several methodological approaches–for example:

i. Collecting longitudinal data over a time period (e.g., 14 years), consisting of four time-points–for example: 10–12 yrs., 14–16 yrs., 18–20 yrs., and 22–24 yrs.

ii. Collecting cross-sectional data, consisting of four distinct cohorts–for example: Developmental Period A (e.g., 10–12 yrs.), Developmental Period B (e.g., 14–16 yrs.), Developmental Period C (e.g., 18–20 yrs.), and Developmental Period D (e.g., 22–24 yrs.).

A development approach (e.g., using cross-sectional data, which may consist of distinct cohorts) may provide theoretical insights into the 'critical period' or periods of the optimization of optimal best practice [1–3]. One possibility may involve some form of comparative analysis of mean scores of flow states between periods–for example: flow state for Period A

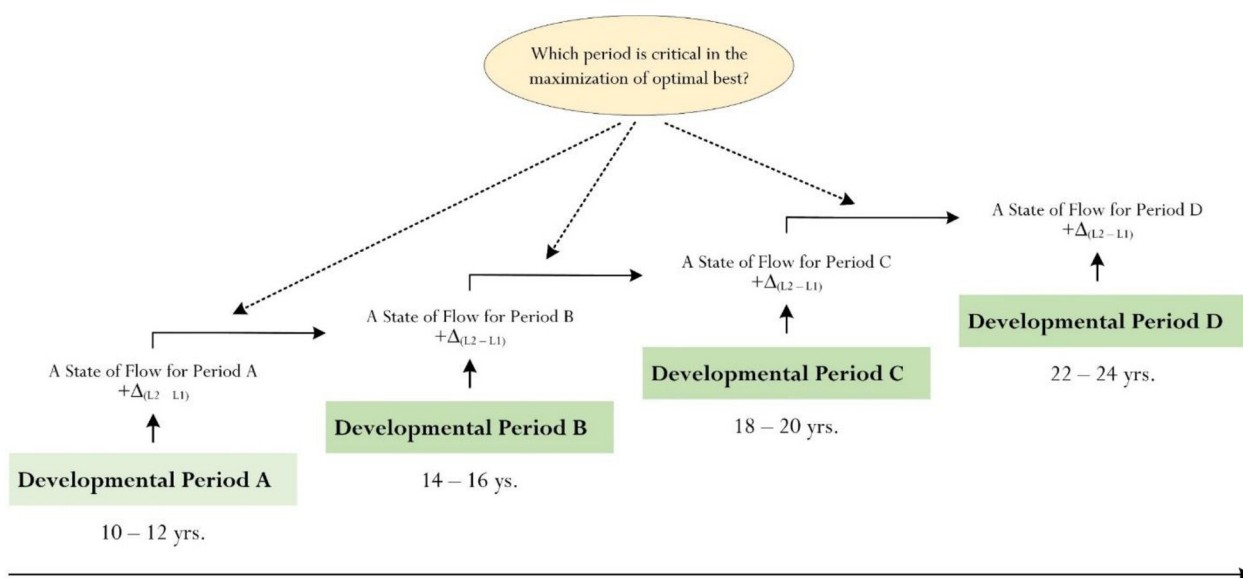

**Fig 4. Optimal best practice 'Window' for consideration.**

*versus* flow state for Period B, flow state for Period B *versus* flow state for Period C, etc. Which period has the highest mean score and why is this the case? Does a high mean score for a particular period (e.g., say Period C in Fig 4) have any meaning that may relate to a critical period of optimal best practice? Can we definitively conclude that a high mean score for a particular period is associated with factors that may account for and/or explain the importance of a critical period?

It is a noteworthy feat to consider and/or to explore a critical period, which may define and assist in the maximization of optimal best practice. Such focus of inquiry, we contend, could advance knowledge and theoretical understanding of a person's motivational mindset. For example, when is it best to encourage, assist, promote, and/or cultivate optimal best practice? Is there a particular period in life when a person has an appropriate 'level of temperament' to *maximize* his optimal best experiences in life? In a similar vein, is it plausible to infer that there is a particular period in life when one is most conducive and/or willing for changes? These posed questions offer an alternative or a contradictory viewpoint to that of flow [22, 82, 83], which emphasizes the *importance of contextualization* for maximized flow experiences (e.g., playing an online computer game).

We acknowledge that our proposition of a critical period or a window for optimal best practice may not have any logical merit for further advancement. By all account, it may well be that optimization of best practice [1, 18] is simply contextualized and personal, reflecting individuals' motivational beliefs, previous experiences, states of mental resolute, etc. Existing writings [1, 18] have focused on this narrative, suggesting that optimization of best practice is context-dependent, requiring expenditure of educational, psychological, and/or psychosocial resources.

## Conclusions

Overall, then, the present study has provided robust evidence that supported our original *a priori* model, which advances the study of the underlying nature of optimal best practice [1–3].

Fundamental to our proposition, as supported by previous theorizations (e.g., Fig 1) [1, 21], is the central position or placement of optimal best practice–namely, that it could serve as an outcome (e.g., academic self-concept → optimal best practice) as well as a predictor (e.g., optimal best practice → academic engagement) and potential mediator of future adaptive outcomes (e.g., optimism → optimal best practice → academic engagement). Moreover, our proposition considers an interesting viewpoint, which delves into the importance of a person's motivational mindset and how this could help facilitate and/or account for his achievement of optimal best practice. For example, from our point of view, a positive and motivational mindset (e.g., feeling optimistic, buoyant, and resolute) would assist to facilitate a person's achievement of optimal best practice.

Evidence derived from data collected in Spain is clear and consistent with those findings established in earlier studies. Moreover, aside from clarity and consistency in empirical findings, our study has also added a cultural element, highlighting the importance in cross-cultural comparisons (e.g., Spanish students *versus* Taiwanese students) of students' optimal best experiences in different sociocultural contexts. That optimal best practice would exert a positive effect on different types of adaptive outcomes (e.g., optimal best practice → adaptive outcome) is validated from students' responses, situated in contrasting learning and sociocultural contexts (e.g., Spanish students *versus* Taiwanese students). Such comparable patterns in associations are significant, cross-culturally, and may support for the continuation of existing research development into the nature of optimal best practice [1–3].

Overall, then, as a recap, our path analysis of university students' responses to a suite of Likert-scale questionnaires yielded a number of interesting findings, which supported our original propositions–that: (i) academic self-concept, optimism, and pessimism predicted a state of $L_2$, and that only academic self-concept and optimism predicted a state of $L_1$, (ii) both $L_1$ and $L_2$ positively influenced proactive academic engagement, and (iii) in line with existing research development is the positive association between $L_1$ and $L_2$.

## Acknowledgments

The authors would like to acknowledge and thank for help of the CEINSA, University of Almeria.

## Author Contributions

**Conceptualization:** Huy P. Phan, Bing H. Ngu.

**Data curation:** Antonio Granero-Gallegos.

**Formal analysis:** Antonio Granero-Gallegos.

**Funding acquisition:** Antonio Granero-Gallegos.

**Investigation:** Antonio Granero-Gallegos, Huy P. Phan, Bing H. Ngu.

**Methodology:** Antonio Granero-Gallegos.

**Project administration:** Antonio Granero-Gallegos.

**Supervision:** Huy P. Phan, Bing H. Ngu.

**Validation:** Antonio Granero-Gallegos, Huy P. Phan, Bing H. Ngu.

**Writing – original draft:** Antonio Granero-Gallegos, Huy P. Phan, Bing H. Ngu.

**Writing – review & editing:** Antonio Granero-Gallegos, Huy P. Phan, Bing H. Ngu.

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
