## [Decision Letter · Decision Letter 0]

3 Apr 2023

PONE-D-22-23792Advancing the study of levels of best practice: Associations with both positive and negative achievement-related experiencesPLOS ONE

Dear Dr. Gallegos

Thank you for submitting your manuscript to PLOS ONE. After careful consideration, we feel that it has merit but does not fully meet PLOS ONE’s publication criteria as it currently stands. Therefore, we invite you to submit a revised version of the manuscript that addresses the points raised during the review process.

ACADEMIC EDITOR: Firstly, I would like to congratulate you on making it this far in the review process.Secondly, I commend you for putting together a generally well received manuscript by the reviewers.Please note that the same have requested some minor changes to your submission and I concur with the revision aspects pointed out.The revisions are listed for your attention, towards the end of this letter.Please ensure that you satisfy the reviewer recommendations as accurately as possible and if/where applicable, provide justification for any dissenting views.I also recommend that you have a final editing and typesetting done on your manuscript, where possible, to ensure a swifter publication of your research paper.I wish you all the best with your revisions.

We look forward to receiving your revised manuscript.

Kind regards,

Musawenkosi Donia Saurombe, Ph.D.

Academic Editor

PLOS ONE

Journal Requirements:

2. Peer review at PLOS ONE is not double-blinded (https://journals.plos.org/plosone/s/editorial-and-peer-review-process). For this reason, authors should include in the revised manuscript all the information removed for blind review.

“This research was funded by the XXXXXXXX PLAN FOR RESEARCH, DEVELOPMENT, AND INNOVATION (xxxxxxx; anonymized) OF THE xxxxxxxxxx (anonymized), grant number XXXXXXX (anonymized) (I+D+i research project)”.

““This research was funded by the ANDALUSIAN PLAN FOR RESEARCH, DEVELOPMENT, AND INNOVATION (PAIDI 2020) OF THE JUNTA DE ANDALUCÍA, grant number P20_00148 (I+D+i research project)”.

Reviewers' comments:

Reviewer's Responses to Questions

**Comments to the Author**

1. Is the manuscript technically sound, and do the data support the conclusions?

Reviewer #1: Yes

Reviewer #2: Yes

2. Has the statistical analysis been performed appropriately and rigorously? 

Reviewer #1: Yes

Reviewer #2: Yes

3. Have the authors made all data underlying the findings in their manuscript fully available?

Reviewer #1: Yes

Reviewer #2: Yes

4. Is the manuscript presented in an intelligible fashion and written in standard English?

Reviewer #1: Yes

Reviewer #2: Yes

5. Review Comments to the Author

Reviewer #1: Additional Comments to the Author

Advancing the study of levels of best practice: Associations with both positive and negative achievement-related experiences

• The title seems not to fully reflect the contents of the manuscript (Antecedents � +�(L1 – L2) � Adaptive Outcome) (antecedents as identified in the contents are not reflected in the title). If possible consider adding the context of the study to the title as well.

• Clarifying and positioning the levels of best practice at the beginning of the study will help readers to follow through the study (Actual best practice L1, Optimal best practice L2)

• Figure 2 shows a clear conceptual model of the study, the antecedents are identified, this clarifies the study better however the introduction section should clearly align with the title as well as the conceptual model.

• Indicate how the three comparable and comparative antecedents were identified and why they were selected for this study.

• The data analysis and interpretation were adequately done and present interesting findings.

• Theoretical contributions are clearly outlined; however, the practical contributions are not fully presented.

• Align the conclusions to the three major tenets identified under the section conceptualization of a Theoretical Model for Examination (line 144 to 148). Providing a conclusion for each examined question would shade more light in the summary/conclusion

Reviewer #2: The manuscript is accepted but requires minor changes. The authors should consider avoiding pronouns and write in the third-party writing style. Manuscript needs editing in relation with references, especially intext citation. There should be consistent use of referencing style.

the statistical analysis (i.e., descriptive statistics and correlations) is clear.

6. PLOS authors have the option to publish the peer review history of their article (what does this mean?). If published, this will include your full peer review and any attached files.

Reviewer #1: **Yes: **Dr Martha Harunavamwe

Reviewer #2: No

---

## [Author Response · Author response to Decision Letter 0]

24 May 2023

Manuscript ID

PONE-D-22-23792

We thank the reviewers for his/her constructive comments and his/her thorough revision of the manuscript. Below we answer his/her questions and concerns, including explicitly the changes made in the manuscript as well.

Musawenkosi Donia Saurombe, Ph.D.

Academic Editor

PLOS ONE

Journal Requirements:

• Reponse: Done

2. Peer review at PLOS ONE is not double-blinded (https://journals.plos.org/plosone/s/editorial-and-peer-review-process). For this reason, authors should include in the revised manuscript all the information removed for blind review.

• Reponse: Done

• Reponse: The “written” consent has been specified in the procedure section.

“This research was funded by the XXXXXXXX PLAN FOR RESEARCH, DEVELOPMENT, AND INNOVATION (xxxxxxx; anonymized) OF THE xxxxxxxxxx (anonymized), grant number XXXXXXX (anonymized) (I+D+i research project)”.

““This research was funded by the ANDALUSIAN PLAN FOR RESEARCH, DEVELOPMENT, AND INNOVATION (PAIDI 2020) OF THE JUNTA DE ANDALUCÍA, grant number P20_00148 (I+D+i research project)”.

• Reponse: Information pertaining to our source of funding has been removed from the manuscript, and instead this information has been added in the cover letter (re-submission). 

• Reponse: This information has been included.

• Reponse: No correction is needed from us. 

 

Reviewers' comments:

Reviewer's Responses to Questions

1. Is the manuscript technically sound, and do the data support the conclusions?

Reviewer #1: Yes

Reviewer #2: Yes

2. Has the statistical analysis been performed appropriately and rigorously?

Reviewer #1: Yes

Reviewer #2: Yes

3. Have the authors made all data underlying the findings in their manuscript fully available?

Reviewer #1: Yes

Reviewer #2: Yes

4. Is the manuscript presented in an intelligible fashion and written in standard English?

Reviewer #1: Yes

Reviewer #2: Yes

5. Review Comments to the Author

Additional Comments to the Author

Reviewer – 1

Comment: The title seems not to fully reflect the contents of the manuscript (Antecedents � +�(L1 – L2) � Adaptive Outcome) (antecedents as identified in the contents are not reflected in the title). If possible consider adding the context of the study to the title as well.

• Response: We have revised the title to reflect the context of our study, likewise sample characteristics. The new title is: Advancing the study of levels of best practice of pre-service teacher education students from Spain: Associations with both positive and negative achievement-related experiences

Comment: Clarifying and positioning the levels of best practice at the beginning of the study will help readers to follow through the study (Actual best practice L1, Optimal best practice L2)

• Response: To ensure that we do not repeat ourselves, we have added in one sentence in the introduction to allude readers to the concepts of ‘actual best’ and ‘optimal best’. The sentence is as follows:

“This reflective question, importantly, alludes readers to an important theoretical premise, namely, the notation of ‘�(L1 – L2)’, which depicts the relationship between ‘actual best’, denoted as L1, and ‘optimal best’, denoted as L2.”

Comment: Figure 2 shows a clear conceptual model of the study, the antecedents are identified, this clarifies the study better however the introduction section should clearly align with the title as well as the conceptual model.

• Response: To ensure that we do not repeat ourselves, we have added in two sentences to recap the focus of inquiry, which closely aligns to our conceptualization (i.e., Fig 2) in the Introduction Section. The sentences are: 

“Overall, then, we conceptualize a theoretical model for examination, which seeks to explore and identify different types of antecedents that could impart contrasting effects on the two levels of best practice. Furthermore, as described earlier, a related issue for examination is the explanatory and predictive role of the two levels of best practice.”

Comment: Indicate how the three comparable and comparative antecedents were identified and why they were selected for this study.

• Response: We have now added in a paragraph to indicate why we have chosen self-concept, optimism, and pessimism as antecedents for examination. The paragraph is as follows:

“An examination of the educational psychology literature indicates that there are numerous psychological variables (e.g., self-efficacy for learning)[1-3] and psychosocial factors (e.g., the impact of the home environment)[4, 5], which could function as antecedents of future adaptive outcomes [6-9]. By all account, choosing which psychological variable and/or psychosocial factor to examine is a personal feat, coinciding with existing research development, theoretical premise(s), a researcher’s interest, etc. Our choosing of three psychological variables (i.e., self-concept, optimism, and pessimism) for undertaking, in this case, is based on three important factors: (i) clear and consistent evidence from existing research development, which indicates the predictive role of the psychological variable (e.g., the predictive effect of optimism on academic performance)[10], (ii) advancing knowledge and theoretical understanding of the theoretical premise of the process of human optimization [11-13], which contends that different types of psychological variables may help to optimize a person’s state of functioning, and (iii) intellectual curiosity to seek theoretical understanding into the comparative antecedent roles of the chosen psychological variables (e.g., optimism versus pessimism)”. 

Comment: The data analysis and interpretation were adequately done and present interesting findings.

• Response: Thank you very much.

Comment: Theoretical contributions are clearly outlined; however, the practical contributions are not fully presented.

• Response: We have revised the manuscript to ensure that the practical contributions section is more explanatory. This refinement included two sub-headings, titled:

i. Educational Implications for Consideration

ii. Methodological Insights for Development

Furthermore, we have expanded the Educational Implications for Consideration Section to included the following:

“As conceptualized in Fig 2, optimal best practice may operate as a central variable, which in turn would mobilize, govern, and/or direct a person’s thought and behavioural patterns for effective and/or non-learning experiences. With this in mind, it is plausible for us to consider the following for educational (or non-educational) purposes:

i. Optimal best practice as an important source of information, which could operate to optimize and motivate a person’s thought and behavioural patterns [1-3]. This considered viewpoint contends that perhaps, one (e.g., a teacher) could encourage individuals to focus on striving to achieve optimal best practice. Optimal best practice is personal and subject to individual variations between individuals, reflecting their differing levels of optimism, motivation, aspiration, positive outlook, etc. Striving to achieve optimal best practice, which is a personal best goal for accomplishment [8], may help motivate individuals to engage in the learning and/or non-learning processes. Within the context of academic learning in a specific subject or a variety of subjects, it is plausible for educators to use a psychological concept known as personal resolve [5, 6, 21], which we cited earlier, to encourage, promote, and/or facilitate optimal best practice. Personal resolve, in brief, is defined as a person’s “internal state of decisiveness and resolute to strive for optimal achievement best in an optimistic manner” [84]. As such, from this definition, personal resolve emphasizes the importance of a student’s self-determination to overcome any obstacles that may arise, guiding and directing her to engage in a purposive act to achieve optimal best (i.e., personal resolve � L2)[21, 84]. In this sense, personal resolve is analogous to an internal state of mental fortitude, serving as a potent source of motivation for learning and non-learning purposes. Timely use of positive feedback in class, for example, may instil and/or encourage a heightened state of personal resolve, which in turn could motivate a student to strive for optimal best practice. 

ii. Design pathways, means, and/or opportunities, which could stimulate motivation for achievement of optimal best practice. Our present findings, for example, indicate support for the potential use of academic self-concept [e.g., 35, 36, 37], optimism [e.g., 33, 42, 43, 44], and other positive constructs (e.g., hope) [85-87] that could, indeed, serve to encourage and/or to stimulate the striving of optimal best. For example, positive verbal discourse (e.g., encouraging feedback, which may reflect a state of hope and inspiration) may instil a level of academic self-concept for learning, in turn helping to motivate students to strive for optimal best experiences. In a similar vein, encouraging a student to show optimism regardless of perceived difficulties, hardships, etc. could also help.

Interesting subject contents and/or contextual matters that have daily relevance, likewise, may assist to encourage individuals to strive to achieve optimal best practice. In terms of academic learning, for example, interesting subject contents (e.g., the study of paranormal psychology)[88] may encourage situational interest, intellectual curiosity, and aspiration for mastery, all of which could motivate a student to strive to achieve optimal best experiences. Difficult and/or uninterest subject contents (e.g., the learning of algebra)[20] that are perceived as being ‘boring’ [89], in contrast, are more likely to demotivate, weakening one’s interest, intellectual curiosity, aspiration to strive to achieve mastery. In a similar vein, a contextual matter or situation that has life relevance and/or daily applicability (e.g., the practice of meditation)[90-92] is more likely to enhance, stimulate, and/or facilitate, motivating and guiding a person to strive for optimal life functioning”. 

Comment: Align the conclusions to the three major tenets identified under the section conceptualization of a Theoretical Model for Examination (line 144 to 148). Providing a conclusion for each examined question would shade more light in the summary/conclusión.

• Response: We have now added a final paragraph to conclude our article. The paragraph, in particular, summarizes our study and its results in light of the three tenets raised in the conceptualization. The concluding paragraph is as follows:

“Overall, then, as a recap, our path analysis of university students’ responses to a suite of Likert-scale questionnaires yielded a number of interesting findings, which supported our original propositions – that: (i) academic self-concept, optimism, and pessimism predicted a state of L2, and that only academic self-concept and optimism predicted a state of L1, (ii) both L1 and L2 positively influenced proactive academic engagement, and (iii) in line with existing research development is the positive association between L1 and L2”. 

Reviewer - 2

Comment: The manuscript is accepted but requires minor changes. The authors should consider avoiding pronouns and write in the third-party writing style. Manuscript needs editing in relation with references, especially intext citation. There should be consistent use of referencing style. The statistical analysis (i.e., descriptive statistics and correlations) is clear.

• Response: We have gone back and ensured that our in-text referencing, using PLoS One’s Referencing style, is correct. We also thank Reviewer 2 for his/her suggestion regarding third-person. However, in recent years, the APA 7th Edition Manual has recommended the use of first-person; first-person, according to the APA Manual, is more engaging. 

References

1. Bandura A. Self-efficacy: Toward a unifying theory of behavioral change. Psychological Review. 1977;84(2):191-215. PubMed PMID: 847061.

2. Bandura A. Self-efficacy: The exercise of control. New York: W. H. Freeman & Co; 1997.

3. Bandura A. Self-efficacy in changing societies: Cambridge university press; 1995.

4. McCartney K, Dearing E, Taylor BA, Bub KL. Quality child care supports the achievement of low-income children: Direct and indirect pathways through caregiving and the home environment. Journal of Applied Developmental Psychology. 2007;28(5-6):411-26. doi: 10.1016/j.appdev.2007.06.010. PubMed PMID: WOS:000250159800005.

5. Muola JM. A study of the relationship between academic achievement motivation and home environment among standard eight pupils. Educational Research and Reviews. 2010;5(5):213-7.

6. Phan HP, Ngu BH. Teaching, Learning and Psychology. Docklands, Melbourne: Oxford University Press; 2019 10th November, 2015.

7. Eggen P, Kauchak D. Educational psychology: Windows on classrooms. 9th ed. Harlow, Essex: Pearson Education Limited; 2014.

8. Moreno R. Educational psychology. Hoboken, NJ: John Wiley & Sons, Inc.; 2010.

9. Ormrod JE. Educational psychology: Developing learners. 5th ed. Upper Saddle River, NJ: Pearson Merrill Prentice Hall; 2006.

10. Wu JH. Academic optimism and collective responsibility: An organizational model of the dynamics of student achievement. Asia Pacific Education Review. 2013;14(3):419-33. doi: 10.1007/s12564-013-9269-6. PubMed PMID: WOS:000323652700014.

11. Fraillon J. Measuring student well-being in the context of Australian schooling: Discussion Paper. Carlton South, Victoria: The Australian Council for Research; 2004.

12. Phan HP, Ngu BH, Yeung AS. Achieving optimal best: Instructional efficiency and the use of cognitive load theory in mathematical problem solving. Educational Psychology Review. 2017;29(4):667-92. doi: http://10.1007/s10648-016-9373-3. PubMed PMID: WOS:000414547000001.

13. Phan HP, Ngu BH, Yeung AS. Optimization: In-depth examination and proposition. Frontiers in Psychology. 2019;10(Article 1398):1-16. doi: 10.3389/fpsyg.2019.01398.

---

## [Editor Report · Decision Letter 1]

15 Jun 2023

Advancing the study of levels of best practice: Associations with both positive and negative achievement-related experiences

PONE-D-22-23792R1

Dear Dr. Granero-Gallegos,

We’re pleased to inform you that your manuscript has been judged scientifically suitable for publication and will be formally accepted for publication once it meets all outstanding technical requirements.

Within one week, you’ll receive an e-mail detailing the required amendments. When these have been addressed, you’ll receive a formal acceptance letter, and your manuscript will be scheduled for publication.

An invoice for payment will follow shortly after the formal acceptance. To ensure an efficient process, please log into Editorial Manager at http://www.editorialmanager.com/pone/, click the 'Update My Information' link at the top of the page, and double check that your user information is up to date. If you have any billing related questions, please contact our Author Billing department directly at authorbilling@plos.org.

Kind regards,

Musawenkosi Donia Saurombe, Ph.D.

Academic Editor

PLOS ONE

Additional Editor Comments (optional):

I would like to commend the author(s) on their diligent effort in revising the manuscript, as per the reviewer comments and feedback. I was pleased with the responses and the manner in which all reviewer recommendations were addressed - well done and congratulations!
---

## [Editor Report · Acceptance letter]

23 Jun 2023

PONE-D-22-23792R1 

Advancing the study of levels of best practice pre-service teacher education students from Spain: Associations with both positive and negative achievement-related experiences 

Dear Dr. Granero-Gallegos:

I'm pleased to inform you that your manuscript has been deemed suitable for publication in PLOS ONE. Congratulations! Your manuscript is now with our production department. 

Kind regards, 

on behalf of

Prof. Musawenkosi Donia Saurombe 

Academic Editor

PLOS ONE